# Indoor Carbon Dioxide, Fine Particulate Matter and Total Volatile Organic Compounds in Private Healthcare and Elderly Care Facilities

**DOI:** 10.3390/toxics10030136

**Published:** 2022-03-12

**Authors:** Alexandre Baudet, Estelle Baurès, Olivier Blanchard, Pierre Le Cann, Jean-Pierre Gangneux, Arnaud Florentin

**Affiliations:** 1Faculté d’Odontologie, CHRU-Nancy, Université de Lorraine, F-54505 Vandoeuvre-les-Nancy, France; 2APEMAC, Université de Lorraine, F-54000 Nancy, France; arnaud.florentin@univ-lorraine.fr; 3EHESP School of Public Health, Inserm, IRSET (Institut de Recherche en Santé, Environnement et Travail)—UMR_S 1085, Université de Rennes, F-35000 Rennes, France; estelle.baures@ehesp.fr (E.B.); olivier.blanchard@ehesp.fr (O.B.); pierre.lecann@ehesp.fr (P.L.C.); jean-pierre.gangneux@univ-rennes1.fr (J.-P.G.); 4Laboratoire de Parasitologie-Mycologie, CHU-Rennes, F-35000 Rennes, France; 5Faculté de Médecine, CHRU-Nancy, Université de Lorraine, F-54505 Vandoeuvre-les-Nancy, France

**Keywords:** indoor air quality, carbon dioxide, particulate matter, total volatile organic compounds, dental offices, general practitioner offices, pharmacies, nursing homes

## Abstract

Poor indoor air quality can have adverse effects on human health, especially in susceptible populations. The aim of this study was to measure the concentrations of dioxide carbon (CO_2_), fine particulate matter (PM_2.5_) and total volatile organic compounds (TVOCs) in situ in private healthcare and elderly care facilities. These pollutants were continuously measured in two rooms of six private healthcare facilities (general practitioner’s offices, dental offices and pharmacies) and four elderly care facilities (nursing homes) in two French urban areas during two seasons: summer and winter. The mean CO_2_ concentrations ranged from 764 ± 443 ppm in dental offices to 624 ± 198 ppm in elderly care facilities. The mean PM_2.5_ concentrations ranged from 13.4 ± 14.4 µg/m^3^ in dental offices to 5.7 ± 4.8 µg/m^3^ in general practitioner offices. The mean TVOC concentrations ranged from 700 ± 641 ppb in dental offices to 143 ± 239 ppb in general practitioner offices. Dental offices presented higher levels of indoor air pollutants, associated with the dental activities. Increasing the ventilation of these facilities by opening a window is probably an appropriate method for reducing pollutant concentrations and maintaining good indoor air quality.

## 1. Introduction

The indoor air quality (IAQ) is a major concern because air pollutants may pose health risks and comfort problems [1]. The IAQ in healthcare and elderly care facilities is particularly important because these facilities receive both workers and the public, including susceptible populations such as the elderly, patients with chronic respiratory diseases and immunocompromised patients. In Europe, workers spend more than 30% of their time working indoors [2], and the elderly are widely exposed to indoor air because they spend more than 80% of their time indoors at home [3]. Thus, respiratory diseases among the elderly living in care facilities have been reported due to their exposure to indoor air pollutants [4]. In the last decade, the IAQ has been widely studied, and several European countries have started to develop guidelines and specific legislation regarding air quality in closed workplaces. Supplementary studies on IAQ are required to improve the scientific knowledge, and to define guide values, reference values and action values for IAQ [5].

The IAQ is a multi-disciplinary issue which is determined by many pathways, including chemical pollutants (such as carbon dioxide [CO_2_], volatile organic compounds [VOCs], semi-volatile organic compounds [SVOCs], particulate matter [PM]) and biological contaminants (bacteria, fungi, virus and pollen). To monitor the IAQ and to easily identify poor IAQ, Wong et al. demonstrated the effectiveness of a set of three parameters: CO_2_, respirable suspended particulates and total volatile organic compounds (TVOCs) as an indicator for the office IAQ [6].

CO_2_ concentration is well known as a proxy for IAQ, as well as a risk marker for transmission of airborne diseases [7,8]. It is an important indicator of ventilation (fresh air supply) in occupied indoor environments [9]. CO_2_ is a byproduct of human metabolism and exists in high concentrations in exhaled air (approximately 40,000 ppm) compared with outdoor air (approximately 400 ppm) [10]. A room presenting a CO_2_ concentration above 1000 ppm is considered an indicator of unacceptable ventilation and can negatively affect people’s perceptions and performance [11]. Finally, indoor concentration of CO_2_ is now used as an indicator in the context of the COVID-19 pandemic by the Centers for Disease Control and Prevention (CDC) and by the French Public Health Council, for example [12,13].

Concerning respirable particulates, the concentrations of fine particulates (PM with aerodynamic diameter < 2.5 μm [PM_2.5_]) in the air are interesting from a health point of view because they are capable of penetrating deep inside the lungs [14]. PM_2.5_ is a robust indicator of the risk associated with a mixture of pollutants from ambient air pollution. Worldwide, the exposure to PM_2.5_ is identified as a major risk leading to lower life expectancy and/or lives with disease [15]. Long- and short-term PM_2.5_ exposure increases morbidity (especially respiratory and cardiovascular diseases) and mortality [1]. These health effects are influenced by PM_2.5_ concentration, components, size, exposure time to PM_2.5_ and human age [16].

TVOCs may be used as an indicator for chemical load in the indoor environment, and may reveal the insufficient ventilation in a room [11]. The work environment may be an important setting for exposure to VOCs [17]. It is well known that healthcare workers are exposed to complex mixtures of chemicals, partly due to cleaning and disinfecting products widely use in these environments [18,19,20]. Continuous measurements of TVOCs are interesting to obtain an overview of chemical pollution and to describe short-duration mean or peak exposures that may be particularly relevant for asthma and irritation symptoms [18].

Improvements in IAQ in settings such as healthcare and care facilities are become a major concern. In these environments, the limitations of CO_2_, PM_2.5_ and TVOCs are mainly ensured by frequent changes in the air with natural, and sometimes mechanical ventilation. The IAQ has been poorly studied in private healthcare and elderly care facilities, especially before specific ventilation recommendations in these facilities since the COVID-19 pandemic. Knowledge of the pollutant levels in the indoor air of these facilities in normal working conditions (without specific measures due to the COVID-19 context) is important to evaluate the health risk for exposed people. Moreover, the spatiotemporal monitoring of IAQ by key indicators may provide information about emissions sources, ventilation and subsequent personal exposure [21,22].

The main aim of this study was to measure the concentrations of CO_2_, PM_2.5_ and TVOCs in the indoor environments of private healthcare and elderly care facilities in 2018/2019. The secondary aim was to describe the diurnal and seasonal (summer/winter) variations in these indoor air pollutants.

## 2. Materials and Methods

### 2.1. Study Campaigns

The study was carried out in two French urban areas (Nancy and Rennes). In each urban area, the measurements were performed in five facilities: three private healthcare facilities (dental office, general practitioner office and pharmacy) and two elderly care facilities (nursing homes).

The study was carried out in two seasons: summer and winter. The summer campaign was conducted in June 2018 in Rennes and in June 2019 in Nancy. The winter campaign was conducted in February 2019 in Rennes and in Nancy.

### 2.2. Facilities Characteristics

The characteristics of the buildings and sampled rooms have been described in a previous article [19]. Regarding the ventilation characteristics, all elderly care facilities had mechanical ventilation (4/4), and 50% of dental offices (1/2) were fitted. The air change rate of mechanical ventilation was calculated in each room using the volume of the room measured with a laser rangefinder, and the flow rate of each air vent of mechanical ventilation measured once during the study with Q-Trak^®^ 7565 (TSI Inc., Shoreview, MN, USA) (Figure 1A). In the elderly care facilities, the air change rates of mechanical ventilation were 0.4 ± 0.4 Vol/h (range: 0.1–1.0) and 0.4 ± 0.3 Vol/h (range: 0.2–1.0) in common rooms and bedrooms, respectively. In the fitted dental office, they were 0.3 and 0.8 Vol/h in treatment room and waiting room, respectively. General practitioner offices and pharmacies used only natural ventilation. All the studied rooms had exterior walls with openable windows, except for a pharmacy in a shopping arcade and the sterilization room of a dental office.

The activities performed and the number of persons in the rooms were not recorded in real time during the study.

The median temperature/relative humidity—measured using Class’Air^®^ (PYRESCOM, Canohès, France)—were 24.9 °C/47.4% (range: 20.6–28.9 °C/32.0–68.0%) during the summer and 20.8 °C/37.2% (range: 13.3–26.6 °C/22.0–59.0%) during the winter.

### 2.3. Data Collection

In each facility, ambient concentrations of CO_2_, PM_2.5_ and TVOCs were measured in two rooms: the treatment room and waiting or sterilization room in dental offices, the consulting room and waiting room in general practitioner offices, commercial space and storage room in pharmacies, and the common room (refectory or lounge) and bedroom in elderly care facilities.

The measurement equipment was installed at least one meter away from the floor (Figure 1B). Areas directly exposed to ventilation (near to doors and windows) were avoided.

#### 2.3.1. CO_2_ Measurements

The CO_2_ concentration was continuously measured in each room at intervals of 10 min over a period of four and a half days (Monday morning to Friday midday) per campaign using Class’Air^®^ (PYRESCOM, Canohès, France).

#### 2.3.2. PM_2.5_ Measurements

The PM_2.5_ concentration was continuously measured in each room at intervals of 1 min over two-and-a-quarter days (allowing measurements in the two sampled rooms of a facility per week) using an optic particle counter: pDR1500^®^ (Thermo Fisher Scientific Inc., Waltham, MA, USA). The pDR1500^®^ apparatus had a flow rate of 1.5 L/min and was filled using its Blue Cyclone inlet for PM_2.5_. Calibration of the particles counter was performed by the manufacturer before each seasonal campaign.

#### 2.3.3. TVOC Measurements

The TVOC concentration was continuously measured in each room at intervals of 1 min over two-and-a-quarter days (allowing measurements in the two sampled rooms of a facility per week) using a photoionization detector: ppbRAE3000^®^ (RAE Systems Inc., San Jose, CA, USA). Following the manufacturer’s recommendations, two-point calibration of the detector was performed by the sampling team with activated carbon for zero calibration and with isobutylene for span calibration. The calibration was performed in each sampled room before the measurements.

### 2.4. Statistical Analysis

Statistical analysis was performed using RStudio^®^ (RStudio Inc., Boston, MA, USA) version 1.1.456. Data were described as mean ± standard deviation (SD) and range (minimum–maximum). The results were statistically processed with Student’s *t*-tests to compare two datasets, and with ANOVA tests to compare more than two datasets, then with post hoc *t*-tests using the Holm correction. Pearson’s correlation analyses were used to explore the relationships between concentrations of CO_2_, PM_2.5_ and TVOCs. The statistical significance was set at *p* < 0.05.

For statistical analysis, the occupancy hours have been approximated as follows: opening hours of private healthcare facilities (08:00–12:00 and 14:00–19:00), mealtimes in elderly care facilities’ refectories (8:00–10:00, 12:00–14:00 and 18:00–20:00), and nocturnal time in elderly care facilities’ bedrooms (20:00–08:00).

## 3. Results

The mean concentrations of CO_2_, PM_2.5_ and TVOCs were 764 ± 443 ppm (range: 332–2455), 13.4 ± 14.4 µg/m^3^ (range: 2.1–668.5) and 700 ± 641 ppb (range: 0–3290) in dental offices, 657 ± 421 ppm (range: 356–3633), 5.7 ± 4.8 µg/m^3^ (range: 0.6–45.0) and 143 ± 239 ppb (range: 0–3460) in general practitioner offices, 641 ± 182 ppm (range: 384–1259), 9.7 ± 7.4 µg/m^3^ (range: 0.4–259.5) and 233 ± 199 ppb (range: 0–8398) in pharmacies, and 624 ± 198 ppm (range: 334–1871), 9.2 ± 10.7 µg/m^3^ (range: 1.1–245.1) and 149 ± 210 ppb (range: 0–2600) in elderly care facilities, respectively. The dental offices presented higher concentrations of CO_2_, PM_2.5_ and TVOCs than the other facilities (*p* < 0.0001).

There were significant correlations between CO_2_ and TVOC concentrations (r = 0.57, *p* < 0.0001), between CO_2_ and PM_2.5_ concentrations (r = 0.10, *p* < 0.0001), and between PM_2.5_ and TVOC concentrations (r = 0.10, *p* < 0.0001).

Table 1 shows the concentrations of CO_2_, PM_2.5_ and TVOCs in the rooms of private healthcare and elderly care facilities measured during occupancy hours and non-occupancy hours. Higher mean concentrations of CO_2_, PM_2.5_ and TVOCs were measured in the dental sterilization room during occupancy hours (1057 ± 478 ppm, 18.9 ± 26.2 µg/m^3^ and 1274 ± 504 ppb, respectively). Almost all the rooms presented higher concentrations during occupancy hours compared with non-occupancy hours. The concentrations varied significantly between occupancy hours and non-occupancy hours in all the rooms (*p* < 0.0001), except for TVOCs in the bedrooms of elderly care facilities (*p* = 0.6) and for PM_2.5_ in the storage rooms of pharmacies (*p* = 1).

The CO_2_ concentration was strongly correlated to the human presence in the rooms: it increased during occupancy hours and decreased when people lived in the rooms (Figure 2). The concentrations of PM_2.5_ and TVOCs varied in each room according to the human presence and their activities.

Table 2 shows concentrations of CO_2_, PM_2.5_ and TVOCs in the rooms of private healthcare and elderly care facilities according to the season. Higher mean concentrations of CO_2_ were measured in general practitioner consulting rooms in winter (953 ± 663 ppm) and in the dental sterilization room in summer (940 ± 475 ppm). Higher mean concentrations of PM_2.5_ were measured in the dental waiting room in winter (23.6 ± 22.3 µg/m^3^) and in the bedrooms of elderly care facilities in summer (14.5 ± 15.3 µg/m^3^). Higher mean concentrations of TVOCs were measured in the dental sterilization room in winter (789 ± 323 ppb) and in summer (1432 ± 463 ppb). The concentrations varied significantly between summer and winter in all the rooms (*p* < 0.0001), except for CO_2_ in the common rooms of elderly care facilities (*p* = 0.03) and in the commercial spaces of pharmacies (*p* = 1).

## 4. Discussion

### 4.1. CO_2_ Concentrations

The target CO_2_ level for good ventilation is below 800 ppm, following recommendations of the CDC and the French Public Health Council since the COVID-19 pandemic [12,13]. In this study, the mean concentrations only exceeded this target level during the opening hours: in the dental sterilization room (1057 ± 478 ppm), in dental treatment rooms (1016 ± 510 ppm) and in general practitioner consulting rooms (905 ± 592 ppm). In all the studied rooms, the maximal concentration of CO_2_ measured exceeded 1000 ppm and peaks exceeding 2000 ppm were observed in dental and general practitioner offices during opening hours. According to the CO_2_ guidelines set by the German Committee on indoor guide values: concentrations below 1000 ppm are regarded as harmless, those between 1000 and 2000 ppm are elevated (ventilation must be improved) and those above 2000 ppm are unacceptable [23]. Poor ventilation contributes to a high indoor CO_2_ concentration [24].

The mean concentrations of CO_2_ in this study appear similar to those measured in other elderly care facilities (572.1 ± 106.0 ppm) and in different hospitals (649.1 ± 178.6 and 670 ± 190 ppm) [24,25]. CO_2_ is emitted by human respiration, and its concentration is higher in indoor environments with lots of people, such as schools (mean concentrations from 653 ± 54 to 1352 ± 940 ppm) [26].

The dental and general practitioner offices studied presented particularly poor ventilation; however, this study was performed before the COVID-19 pandemic and a lot of recommendations regarding ventilation in healthcare facilities have been published since the pandemic [27]. Regarding dental treatment rooms, other studies performed before the COVID-19 pandemic presented similar results (1265.1 ± 183.2 ppm during opening hours and 618.7 ± 69.8 ppm during non-opening hours) [28], whereas studies performed since the COVID-19 pandemic presented lower CO_2_ concentrations. Huang et al. showed that the CO_2_ concentrations in early morning (5:00–7:00 a.m.) were 421 ± 10 ppm, similar to outdoor levels, and increased during opening hours to a plateau level of 786 ± 207 ppm [7]. Tzoutzas et al. found that only a few outlier values of CO_2_ (<1% of the data) exceeded 1000 ppm [29]. Future investigations are required to confirm that the recent recommendations have improved ventilation with a reduction in the CO_2_ concentration in dental offices, but also in other healthcare and care facilities.

### 4.2. PM_2.5_ Concentrations

The World Health Organization (WHO) has recently revised its air quality guidelines, with a reduction in the recommended levels of long-term exposure to PM_2.5_ from 10 to 5 μg/m^3^ [1]. Except in the consulting rooms of general practitioner offices, the mean concentrations of PM_2.5_ in private healthcare and elderly care facilities were all higher than this new target level: both during occupational hours and non-occupational hours. Thus, efforts to protect populations who work in these facilities are required to reduce the health risks posed by this pollution because PM_2.5_ increases morbidity and mortality from cardiovascular and respiratory disease [30]. Concerning patients’ exposure in healthcare facilities, they are only exposed for a short period of time. The 24 h exposure recommendation from the WHO is 15 μg/m^3^ [1]. Almost all mean concentrations were lower than this target level.

The mean concentration of PM_2.5_ measured in this study appeared to be lower than in French dwellings (37 ± 52 µg/m^3^) [31] but higher than in French hospitals (1.6 µg/m^3^) [20]. PM_2.5_ concentrations and seasonal variations measured in dental offices and pharmacies are similar to those measured by Mandin et al. in building offices, with higher concentrations in winter (15.0 ± 8.4 µg/m^3^) than in summer (9.7 ± 4.6 µg/m^3^) [32]. In hospitals, Baurès et al. found higher concentrations in summer than in winter [20], in accordance with the measurements performed in elderly care facilities in this study.

The spatial and temporal variations in PM_2.5_ concentrations are explained by indoor environmental conditions, such as building characteristics, interior materials, occupant window-opening activities, indoor source emissions and ventilation systems [33,34]. PM_2.5_ concentrations increase when people are present in a room and with human activity, including the resuspension of deposited particles due to human movements [35]. Dental offices presented higher PM_2.5_ concentrations, probably associated with the dental treatments performed which include aerosol-generating procedures [27,29,36]. Indoor PM_2.5_ concentrations are also affected by the infiltration of outdoor PM_2.5_ [29,34], including outside traffic [33,37]. Many of our facilities were in residential neighborhoods close to PM_2.5_ sources such as a car park or major road. 

### 4.3. TVOC Concentrations

The mean concentrations of TVOCs measured in general practitioner offices, pharmacies and elderly care facilities of this study seem lower than those measured in other elderly care facilities [25,38], in hospitals [24], and in primary schools and kindergartens [14]. They seem harmless according to the TVOC guidelines set by the German Committee on Indoor Guide Values because only a few peaks exceeded 500 ppb in these environments. This guideline indicates that TVOC concentrations below 150 ppb (level 1) are safe (target level); those between 150 and 500 ppb (level 2) are regarded as harmless (ventilation is recommended); those between 500 and 1500 ppb (level 3) may induce comfort concerns (contamination sources should be identified and ventilation should be improved); those between 1500 and 5000 ppb (level 4) generate significant comfort issues (exposure should not exceed 1 month, contamination sources should be identified and ventilation should be improved); and those above 5000 ppm (level 5) are unacceptable (such rooms should be used only if unavoidable, but only for short periods (hours) with intensified ventilation) [39,40].

Regarding dental offices, TVOC concentrations were more elevated, especially in treatment and sterilization rooms during opening hours (1059 ± 686 and 1274 ± 504 ppb, respectively), which may induce comfort concerns (level 3). These results are in line with other studies performed in dental clinics which showed very high concentrations of TVOCs during opening hours associated with the nature of the dental activities and ventilation conditions [28,29,36]. Higher concentrations of TVOCs were described in dental treatment rooms during opening hours due to the frequent use of dental materials and biomaterials (such as dentin bonding agents, acrylic resin materials for temporary restorations, various primers and disinfection materials) which may lead to extremely high TVOC emissions [29]. Regarding pharmacies, although mean concentrations appear harmless during opening hours (approximatively 250 ppb), very high peaks were measured (maximal value of 8398 ppb in a storage room and of 3801 ppb in a commercial space).

TVOCs are mainly influenced by indoor environmental factors such as human activities and material use rather than the season, with the emissions of certain VOCs higher during warm seasons [14,19,25]. In healthcare and care facilities, cleaning and disinfecting products—widely performed to reduce the risk of infection—are an important source of VOCs in the indoor air [18,24]. In the facilities investigated in this study, it was previously shown that the main VOCs in indoor air were ethanol and isopropanol (median concentrations of 378.9 and 23.6 μg/m^3^, respectively), which are two alcohols produced by hydro-alcoholic solutions and many disinfectants [19].

### 4.4. Strengths and Limitations of the Study

This study provides new insights regarding the IAQ across three key ambient parameters in private healthcare and elderly care facilities. In addition, measurements were performed in two seasons during several days of typical weeks of activity to consider the diversity of activities in these facilities.

This study presented some limitations. No outdoor measurements were performed to assess the impact of outdoor pollution on indoor parameters, especially indoor/outdoor PM_2.5_ concentrations, which may exhibit a close relationship [26]. The activities performed, the window opening, and the number of persons in the rooms were not recorded in real time. The occupancy hours have been approximated, and some of the results concerning comparisons between occupancy hours and non-occupancy hours should be interpreted with caution. A limited number of facilities were investigated; thus, future investigations are required to confirm the results and characterize indoor air pollution of private healthcare and elderly care facilities more precisely. Future investigations are also required to identify the chemical pollutants related to human activities, daily procedures, internal sources (finishing materials, furniture, products used, etc.), HVAC (heating, ventilation and air conditioning) systems, and their maintenance, cleaning activities and energy performances [41,42].

### 4.5. Recommendations

This study has highlighted that the IAQ should be improved in dental offices because they presented significantly higher concentrations of CO_2_, PM_2.5_ and TVOCs in comparison with other facilities. These concentrations regularly exceeded target levels in dental offices. The studied dental sterilization room presented higher pollutant concentrations during opening hours compared with other studied rooms. The poor IAQ of this room is explained by its poor ventilation: the absence of an opening window and mechanical ventilation. Only opening the door could supply the sterilization room with fresh air from a corridor.

In order to ensure a healthy and comfortable indoor environment, reasonable measures should be used to control indoor concentrations of CO_2_, PM_2.5_ and TVOCs. According to the sources of indoor pollutants, control strategies can be divided into two categories: source control and control during transmission. To improve the IAQ and reduce the pollutant concentrations in indoor environments, the main recommendation is to increase the ventilation by opening windows and/or installing mechanical ventilation [11,16]. The indoor environment significantly contributes to exposure to pollutants, many of which result in internal sources of pollution and have higher concentrations indoors than outdoors [17,43]. For example, due to the persistence of VOCs emitted by cleaning products, and the health risks for exposed people, paying attention to opening windows during and after cleaning activities is especially crucial [26]. Natural ventilation is considered as a healthier ventilation strategy, which is driven by natural forces of wind-driven pressure difference and temperature difference through opening windows and doors [43]. However, when outdoor pollutant concentrations are high (during rush-hour traffic for facilities close to a road, for example), it is recommended to close doors and windows [15]. Mechanical ventilation with good filtration performance or air purifiers was described as a control method in particulate pollution [16,29]. 

Concerning PM_2.5_, it is recommended to perform regular cleaning of the floors and the surfaces to limit the quantity of deposited particles [11], which can be resuspended by human movements [35]. Regarding TVOCs, the use of dental materials in the dental treatment room and the use of cleaning and disinfecting products in all the healthcare and care facilities was a major source of VOCs, but this cannot be reduced. Indeed, cleaning and disinfecting activities are crucial to limit the contamination of healthcare environments and to reduce the risk of healthcare-associated infection [44,45]. However, it is important to close these products when they are not being used to reduce their diffusion into the indoor air.

## 5. Conclusions

The present study confirms that human presence and activities contribute to increasing the concentrations of CO_2_, PM_2.5_ and TVOCs. The concentrations of these three IAQ indicators measured in situ in private healthcare and elderly care facilities was not always in accordance with IAQ guidelines. The higher concentrations in certain rooms, especially in dental offices, indicates a need to develop appropriate control techniques, including increasing the ventilation during occupancy hours with window opening and/or mechanical ventilation.

## Figures and Tables

**Figure 1 toxics-10-00136-f001:**
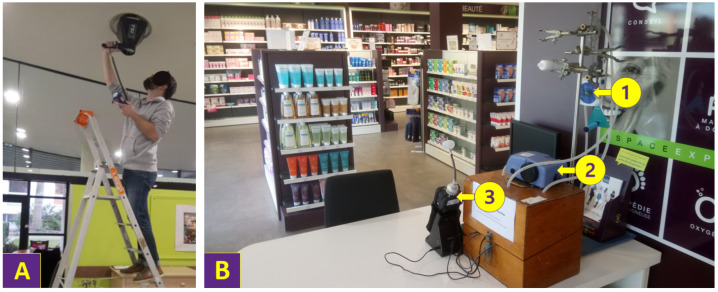
Examples of the flow rate measurements of an air vent in the refectory of an elderly care facility (**A**), and of the CO_2_ (1), PM_2.5_ (2) and TVOC (3) measurements in the commercial space of a pharmacy (**B**).

**Figure 2 toxics-10-00136-f002:**
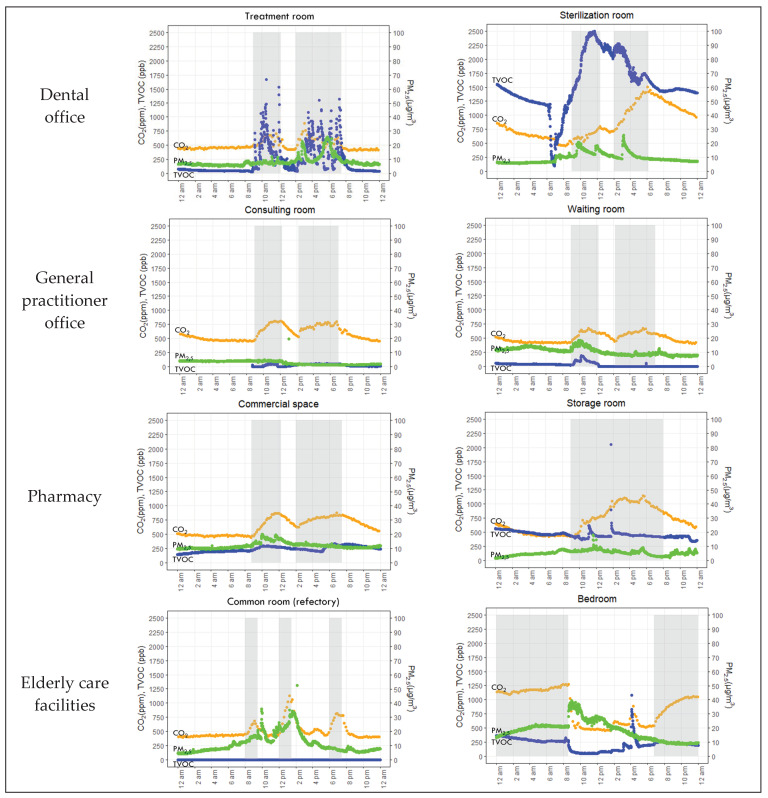
Continuous measurements of CO_2_ (orange), TVOCs (blue) and PM_2.5_ (green) concentrations for 24 h during the summer in different rooms of private healthcare and elderly care facilities. Notes: the shaded periods show the opening hours of healthcare facilities, mealtimes in the refectory and nocturnal time in the bedroom.

**Table 1 toxics-10-00136-t001:** Comparison of mean ± SD (min–max) concentrations of CO_2_, PM_2.5_ and TVOCs measured in different rooms of private healthcare and elderly care facilities according to the occupancy hours (OH) and non-occupancy hours (Non-OH).

Facilities	CO_2_ (ppm)		PM_2.5_ (µg/m^3^)		TVOCs (ppb)	
	OH	Non-OH	OH	Non-OH	OH	Non-OH
Dental offices						
Treatment rooms	1016 ± 510 (378–2455)	676 ± 391 (365–2096)	12.6 ± 7.9 (2.9–101.6)	9.8 ± 8.0 (2.1–126.4)	1059 ± 686 (0–3290)	530 ± 539 (10–1913)
Sterilization room	1057 ± 478 (439–2324)	737 ± 335 (432–2073)	18.9 ± 26.2 (4.7–668.5)	7.5 ± 4.0 (2.2–37.0)	1274 ± 504 (421–3027)	947 ± 467 (95–2259)
Waiting room	542 ± 186 (332–1597)	440 ± 92 (334–990)	16.5 ± 10.5 (4.3–65.6)	19.6 ± 21.1 (5.0–123.7)	193 ± 189 (30–1637)	121 ± 88 (41–734)
General practitioner offices						
Consulting rooms	905 ± 592 (406–3633)	653 ± 403 (406–3373)	4.9 ± 4.7 (0.6–34.9)	3.7 ± 4.0 (0.6–43.0)	200 ± 316 (0–3135)	103 ± 153 (0–2209)
Waiting rooms	676 ± 361 (357–2266)	468 ± 171 (356–1904)	7.6 ± 5.9 (1.1–33.3)	6.5 ± 3.7 (1.3–45.0)	177 ± 236 (0–1783)	96 ± 204 (0–3460)
Pharmacies						
Commercial spaces	717 ± 160 (394–1120)	540 ± 122 (384–1071)	13.3 ± 10.6 (2.9–259.5)	10.6 ± 6.2 (2.8–32.7)	255 ± 162 (0–3801)	227 ± 128 (0–622)
Storage rooms	781 ± 193 (416–1259)	553 ± 119 (392–1127)	7.4 ± 4.2 (0.9–88.8)	7.6 ± 5.1 (0.4–26.8)	241 ± 286 (0–8398)	213 ± 196 (0–2052)
Elderly care facilities						
Common rooms	600 ± 162 (361–1871)	503 ± 144 (334–1843)	8.3 ± 5.3 (1.6–122.6)	6.7 ± 4.0 (1.6–245.1)	146 ± 192 (0–2178)	133 ± 193 (0–2600)
Bedrooms	762 ± 203 (365–1451)	629 ± 174 (345–1322)	9.3 ± 8.9 (1.1–170.0)	12.7 ± 17.6 (1.2–222.0)	160 ± 220 (0–1694)	163 ± 235 (0–2201)

**Table 2 toxics-10-00136-t002:** Comparison of mean ± SD (min–max) concentrations of CO_2_, PM_2.5_ and TVOCs measured in each type of room in private healthcare and elderly care facilities, according to the season.

Facilities	CO_2_ (ppm)		PM_2.5_ (µg/m^3^)		TVOCs (ppb)	
	Summer	Winter	Summer	Winter	Summer	Winter
Dental offices						
Treatment rooms	916 ± 563 (391–2455)	777 ± 391 (365–2169)	10.3 ± 5.2 (3.4–126.4)	12.0 ± 10.1 (2.1–101.6)	807 ± 696 (0–2655)	755 ± 637 (0–3290)
Sterilization room	940 ± 475 (436–2324)	830 ± 389 (432–1876)	9.7 ± 3.1 (5.5–25.7)	16.2 ± 26.4 (2.2–668.5)	1432 ± 463 (95–2598)	789 ± 323 (421–3027)
Waiting room	380 ± 30 (332–568)	602 ± 149 (434–1597)	12.6 ± 2.8 (6.9–28.5)	23.6 ± 22.3 (4.3–123.7)	63 ± 33 (30–549)	252 ± 165 (102–1637)
General practitioner offices						
Consulting rooms	590 ± 164 (432–1105)	953 ± 663 (406–3633)	3.1 ± 1.5 (0.9–19.7)	5.5 ± 5.8 (0.6–43.0)	160 ± 214 (0–2209)	146 ± 283 (0–3135)
Waiting rooms	512 ± 144 (378–943)	602 ± 359 (356–2266)	8.6 ± 5.9 (1.7–33.3)	5.6 ± 3.0 (1.1–45.0)	35 ± 119 (0–2424)	223 ± 256 (0–3460)
Pharmacies						
Commercial spaces	621 ± 148 (393–943)	624 ± 182 (384–1120)	11.0 ± 2.3 (7.6–20.1)	12.4 ± 11.0 (2.8–259.5)	312 ± 104 (5–1340)	192 ± 150 (0–3801)
Storage rooms	690 ± 208 (392–1259)	631 ± 176 (416–1251)	4.0 ± 2.2 (0.4–88.8)	9.7 ± 4.5 (4.7–34.2)	449 ± 80 (263–2052)	84 ± 201 (0–8398)
Elderly care facilities						
Common rooms	523 ± 175 (334–1871)	551 ± 147 (376–1843)	7.7 ± 5.0 (2.0–245.1)	6.8 ± 4.0 (1.6–122.6)	45 ± 155 (0–2600)	248 ± 174 (47–2178)
Bedrooms	604 ± 228 (345–1451)	771 ± 138 (401–1322)	14.5 ± 15.3 (2.1–222.0)	6.9 ± 11.4 (1.1–189.9)	75 ± 130 (0–2201)	313 ± 277 (0–1694)

## Data Availability

The datasets supporting the conclusions of this article are included within the article.

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
