# Peer review of "Indoor Carbon Dioxide, Fine Particulate Matter and Total Volatile Organic Compounds in Private Healthcare and Elderly Care Facilities"

_toxics, 2022, doi:10.3390/toxics10030136_

Round 1
Reviewer 1 Report
This study examined indoor air quality in health facilities. Field measurements of CO2, PM2.5 and TVOC implied the necessity to improve IAQ in these facilities. In general, the paper is well-structured and easy to follow. Some comments for the authors to consider as below:
- Change "ambient" to "indoor" in the title, which better presents the aim and content of the study.
- Line 53 "A room with good ventilation should have CO2 concentration close to 400 ppm" -- the statement is a bit shaky as the CO2 level also depends on occupants' number and activities. Suggest removing.
- Line 56-57, the individual influence of CO2 on human health and comfort is still controversial. CO2 is considered more as an indicator of occupancy and ventilation, whereas the symptoms are probably mainly caused by other chemicals emanated from humans and/or materials. Suggest removing.
- Did the authors measure the ventilation rates continuously? Detailed results of ventilation rate measurement can be provided.
- As the authors define the occupant hours approximately without direct recordings, some of the results concerning comparisons between OH and non-OH hours should be interpreted with caution. For example, the insignificant difference between OH and non-OH for TVOC and PM2.5 in some rooms (Line 169-173) can be simply owing to their distinct occupancy hours from others.
- As the authors did not measure outdoor levels during the campaign, it is too generalized to conclude that opening window is a simple method to maintain good air quality as mentioned in the last sentence of the abstract. Suggest rephrasing.
Reviewer 2 Report
The study addresses in situ measuring of the concentrations of dioxide carbon (CO2), fine particulate matter (PM2.5) and total volatile organic compounds (TVOC) in private healthcare 18 and elderly care facilities.
Overall, the article is well designed, structured and written.
The title is appropriate since it is very illustrative of the study carried out.
The abstract is comprehensive and clear.
The figures are also clear and informative.
References are adequate and updated.
Below are some points that should be improved/justified:
The main issue of the paper is to know the real contribution of the work. In fact, the paper presents a lot of work but no novelty was added. In fact, it is very often found poor IAQ in services buildings, so the results were expectable. Plus, considering the airflow rates presented, poor indoor air quality would be almost guaranteed. Consequently, the main question would be to know the main sources and causes of the problems and/or what should do to solve them. For this purpose, too little information/details were added making it difficult better understand the results. A set of data/details would be important, for instance:
- no information about the methodology to measure the airflow rate is presented; the same methodology was used for natural and mechanical ventilation?
- no details/photos/schemes of the rooms, air vents, measure points, equipment, ... are presented
- "The characteristics of the buildings and sampled rooms are described in a previous article [21]." but, in fact, analysing the reference [21], no concrete details were presented. For instance: which types of natural or mechanical ventilation, which types of filters (mechanical ventilation), whether there is air recirculation (mechanical ventilation), the existence of HVAC systems maintenance plan, occupation rates, control systems, systems’ set points, ...
Based on the above considerations, a strong set of data/recommendations would be welcomed, otherwise, it would be important to justify and emphasize that the key role of the paper is just to raise awareness of the IAQ problem in that type of space/activity.
Reviewer 3 Report
The paper deals an interesting issue, expecially in this period. The IAQ problem is really important, the authors should show better this task starting from the European situation (https://doi.org/10.3390/atmos11040370) and describing a comparison with same or similar situations for better understanding the role played by CO2 (e.g., https://doi.org/10.3390/ijerph17124280 or https://doi.org/10.3390/ijerph17186695). Further, a comparison and some considerations are important for evaluating the limitations and the recommendations reported in this study. Finally, an English revision should be necessary for solving some points where the meaning is obscure.
Round 2
Reviewer 2 Report
The authors had made enough changes and the paper has been improved. I am satisfied with the current version of the paper.
Author Response
Thank you for the review of our manuscript.
Best regards,
Reviewer 3 Report
The authors revised the paper according to the referees' suggestions and now the paper can be considered for publication.
Author Response

(The authors gave the same response as above.)
